# Transcriptome Analysis in Mexican Adults with Acute Lymphoblastic Leukemia

**DOI:** 10.3390/ijms25031750

**Published:** 2024-02-01

**Authors:** Gabriela Marisol Cruz-Miranda, Irma Olarte-Carrillo, Diego Alberto Bárcenas-López, Adolfo Martínez-Tovar, Julian Ramírez-Bello, Christian Omar Ramos-Peñafiel, Anel Irais García-Laguna, Rafael Cerón-Maldonado, Didier May-Hau, Silvia Jiménez-Morales

**Affiliations:** 1Programa de Doctorado, Posgrado en Ciencias Biológicas, Universidad Nacional Autónoma de México, Mexico City 04510, Mexico; gcruz@ciencias.unam.mx (G.M.C.-M.);; 2Laboratorio de Innovación en Medicina de Precisión Núcleo A, Instituto Nacional de Medicina Genómica, Mexico City 14610, Mexico; didier_may@outlook.com; 3Laboratorio de Biología Molecular, Servicio de Hematología, Hospital General de México Dr. Eduardo Liceaga, Mexico City 06720, Mexico; irmaolartec@yahoo.com (I.O.-C.); mtadolfo73@hotmail.com (A.M.-T.);; 4Subdirección de Investigación Clínica, Instituto Nacional de Cardiología Ignacio Chávez, Mexico City 14080, Mexico; 5Departamento de Hematología, Hospital General de México Dr. Eduardo Liceaga, Mexico City 06720, Mexico

**Keywords:** acute lymphoblastic leukemia, adults, transcriptome analysis, mexican patients, microarrays

## Abstract

Acute lymphoblastic leukemia (ALL) represents around 25% of adult acute leukemias. Despite the increasing improvement in the survival rate of ALL patients during the last decade, the heterogeneous clinical and molecular features of this malignancy still represent a major challenge for treatment and achieving better outcomes. To identify aberrantly expressed genes in bone marrow (BM) samples from adults with ALL, transcriptomic analysis was performed using Affymetrix Human Transcriptome Array 2.0 (HTA 2.0). Differentially expressed genes (DEGs) (±2-fold change, *p*-value < 0.05, and FDR < 0.05) were detected using the Transcriptome Analysis Console. Gene Ontology (GO), Database for Annotation, Visualization, and Integrated Discovery (DAVID), and Ingenuity Pathway Analysis (IPA) were employed to identify gene function and define the enriched pathways of DEGs. The protein–protein interactions (PPIs) of DEGs were constructed. A total of 871 genes were differentially expressed, and *DNTT*, *MYB*, *EBF1*, *SOX4*, and *ERG* were the top five up-regulated genes. Meanwhile, the top five down-regulated genes were *PTGS2*, *PPBP*, *ADGRE3*, *LUCAT1*, and *VCAN*. An association between *ERG*, *CDK6*, and *SOX4* expression levels and the probability of relapse and death was observed. Regulation of the immune system, immune response, cellular response to stimulus, as well as apoptosis signaling, inflammation mediated by chemokines and cytokines, and T cell activation were among the most altered biological processes and pathways, respectively. Transcriptome analysis of ALL in adults reveals a group of genes consistently associated with hematological malignancies and underscores their relevance in the development of ALL in adults.

## 1. Introduction

Acute lymphoblastic leukemia (ALL) is a malignant disorder affecting the blood and bone marrow characterized by rapid clinical evolution, biological heterogeneity, and uncontrolled proliferation of lymphoblasts, leading to a progressive loss of differentiation ability [1,2]. This malignancy accounts for about a quarter of acute leukemias in adults and displays high mortality rates, representing an important public health problem [1,2,3,4]. Despite the fact that risk-adapted therapies have improved overall survival (OS) rates in adults with ALL, ranging from 60% to 80%, long-term remission rates in Latin American populations remain low [5,6,7,8]. For instance, studies in Mexico have reported complete remission (CR) rates between 60% and 80% and a 5-year OS of <35% [9,10,11,12,13], which is even worse in the central region. By including adults with ALL from five Mexico City referral Hospitals, The Acute Leukemia Workgroup reported a 3-year OS of 22.1% [12]. Based on the knowledge of childhood ALL, adults receive risk-adapted treatment regimens, drugs directed to specific chimeric genes, and proteins to detect antigens of the tumor cell surface to stimulate antitumor immune response [2,14,15,16]. Nevertheless, adults with ALL are substantially different than children with ALL. Poor prognostic cytogenetic and molecular abnormalities, such as t(9;22) (*BCR::ABL*) and the Ph-like phenotype, are more frequent in adults than in children (15–25% vs. 2–6%, and >25% vs. 10%, respectively), whereas the good prognosis biomarker t(12;22) (*ETV6::RUNX1*) is less frequent in adults (<1% vs. 20–25%, respectively) [17]. By using microarray gene expression, thousands of genes can be surveyed at the same time, and the analysis of tumor tissues may contribute to understanding the biological mechanisms underlying lymphoblast transformation and ALL progression. Additionally, microarray analysis may allow us to identify potential biomarkers with clinical significance, new therapeutic targets, and genes involved in drug resistance and relapse [18,19,20]. In this study, we performed the first transcriptomic analysis in Mexican adults with ALL to identify abnormally expressed genes.

## 2. Results

### 2.1. Features of Patients

To identify an expression signature associated with ALL in adults, we included forty-three patients with the novo ALL and a control group of five healthy subjects. Of the overall cases, 23 (53.5%) were male and 20 (46.5%) were female, with a median age at diagnosis of 33.8 years (range of 18–57 years). Twenty-seven (62.8%) patients were between 18 and <40 years old at the time of diagnosis and were categorized within the adolescents and young adults group, and the remaining 37.2% were >40 years old. Only five (11.6%) patients were classified as having a standard risk. B-lineage ALL was present in 41 (95.3%) cases, while two cases were T-linage. Sixteen patients relapsed, and six of them died. Death was observed in 16 (37.2%) cases during the time of follow-up. The average survival period for our cohort was 353.2 days (ranging from 18 to 661 days) (Table 1).

### 2.2. Assessment of Replicability

To ensure the robustness of our findings and the reliability of the control group, we conducted an in-depth analysis of replicability. Through this analysis, we evaluated the consistency and stability of the control group in comparison with subsets of our patient cohort. We performed a correlation analysis on gene expression profiles between the five control subjects and two randomly divided patient subgroups (Group 1: 21 patients; Group 2: 22 patients). Correlation coefficient analysis demonstrated a strong concordance (correlation coefficient = 0.713) between the control group and both subsets of patients. This analysis provides essential validation for the reliability and adequacy of our control group, ensuring robust and consistent results.

### 2.3. Differential Expressed Genes in Acute Lymphoblastic Leukemia

A supervised hierarchical cluster analysis was performed to compare the gene expression between patients with ALL and the healthy controls. We found 871 DEGs ((FC > 2 or <−2, *p* value < 0.05, FDR < 0.05), of which 781 (95.1%) were coding genes, and the remaining 90 genes were non-coding RNAs. Within the coding genes, 125 were up-regulated, and 656 were down-regulated in cases compared to healthy subjects (Figure 1). The top ten up-regulated genes included *DNTT*, *MYB*, *SOX4*, *EBF1*, *ERG*, *FLT3*, *CD34*, *STMN1*, *CDK6*, and *NAV1*, while the most down-regulated genes were *PTGS2*, *PPBP*, *ADGRE3*, *LUCAT1*, *VCAN*, *TUBB1*, *PF4*, *RGS2*, *SH3BGRL2*, and *CLEC7A* (Table 2).

### 2.4. Altered Pathways and Biological Processes in Acute Lymphoblastic Leukemia

Gene Ontology analysis revealed the DEGs involved in biological processes relevant to the pathophysiology of ALL, such as apoptosis signaling (*p* = 4.22 × 10^−7^), inflammation mediated by chemokine and cytokine signaling (*p* = 2.23 × 10^−5^), toll receptor signaling (*p* = 1.98 × 10^−3^), interleukin signaling (*p* = 1.0 × 10^−3^), and T cell activation (*p* = 3.88 × 10^−2^) as the most affected pathways, whereas IPA analysis confirmed the involvement of DEGs in cancer-associated processes, such as the pathogen-induced cytokine storm signaling pathway (*p* = 1.23 × 10^−13^), crosstalk between dendritic cells and natural killer cells (*p* = 6.71 × 10^−13^), the pyroptosis signaling pathway (*p* = 4.70 × 10^−12^), natural killer cell signaling (*p* = 5.22 × 10^−11^), and phagosome formation (*p* = 5.45 × 10^−11^), which were the most abnormally regulated pathways. Cell-to-cell signaling and interaction, cellular development, cellular growth and proliferation, cell death and survival, and cell signaling were the main molecular and cellular functions (*p* <1.76 × 10^−4^) in which DEGs play a role (Appendix A). Regarding physiological system development and function (*p* < 1.75 × 10^−4^), the main cellular functions were as follows: hematological system development and function, immune cell trafficking, lymphoid tissue structure and development, hematopoiesis, and tissue development. Additionally, coding DEGs were subject to PPI network analysis using the STRING database with confidence over 0.7. We identified 92 nodes with a node degree > 10 (Appendix A). *CD4* ranked first among these genes, followed by *PTPRC*, *FCGR3A*, *TYROBP*, *FCGR3B*, *TLR4*, *TLR2*, and *CCR5*, based on which the gene–gene interactions network was constructed and visualized by using Cytoscape (Figure 2). After completion of the GO function enrichment analysis, these genes were significantly enriched in the KEEG biological immune-related process (Appendix A), and the most affected pathways were inflammation mediated by chemokine and cytokine signaling, apoptosis signaling, interleukin signaling, and Toll receptor signaling pathways.

### 2.5. Validation of DEGs Associated with ALL by Quantitative RT-PCR

RT-PCR was used in a subset of cases and controls to validate the data obtained from microarray analysis. *EBF1* and *FLT3* genes were selected based on previous reports showing abnormal expression in childhood and adulthood ALL [21]. The gene expression of these two genes displays consistent results obtained in microarray expression analysis, and both genes were up-regulated in patients with ALL in contrast to healthy subjects (Appendix A).

### 2.6. Clinical Association and Survival Analysis

OS analysis was performed for the top ten up-regulated and top ten down-regulated genes, revealing that *ERG*, *SOX4*, and *CDK6* were associated with a worse prognosis. The associations between the expression (high or low, according to the median of their expression level) and EFS and OS revealed significant differences in hazard ratios (HRs) (Figure 3A–F). For instance, the high expression of *ERG* (HR = 3.33, 95% CI 1.19–9.34, *p* < 0.021 and HR = 4.24, 95% CI 1.51–15.63, *p* < 0.02) and CDK*6* (HR = 5.33, 95% CI = 1.55–18.35, *p* < 0.007 and HR = 5.6, 95% CI = 1.25–25.44, *p* < 0.024) showed high risk of relapse and death, respectively (Figure 3A–D). Meanwhile, *SOX4* expression was associated with EFS (HR = 2.92 95% CI = 1.068–7.99, *p* < 0.036) but not with OS (HR = 3.37 95% CI = 0.939–12.14, *p* < 0.62) (Figure 3E,F). Genes such as *STMN1* and *CD34* were associated either with EFS or OS, respectively. Regarding the hub genes, low expression of *CD4* was associated with better outcomes (HR = 0.3 95% CI = 0.115–0.79, *p* < 0.0147 and HR = 0.27 95% CI = 0.08–0.87, *p* < 0.089 for EFS and OS, respectively) (Figure 3G,H).

### 2.7. Gene Expression Correlation Analysis

To identify genes with similar profiles related to ALL, gene expression correlation analysis was performed, including a total of 871 DEGs, and three co-expression modules were detected (Figure 4). The largest module comprised 347 genes, and it included five hub genes, as listed in Appendix A.

### 2.8. Potentially Targetable Genes

Based on IPA, we conducted an analysis of genes with potential implications for ALL treatments. This analysis unveiled a comprehensive list of 188 genes with pharmacological relevance. To refine our focus and emphasize the significance of these genes, we implement a stringent selection process. This process led us to identify a subset of nine genes: *FLT3*, *CDK6*, *CD4*, *PTPRC*, *FCGR3A/FCGR3B*, *TLR4*, *CCR7*, and *CD2* (Appendix A). These genes, apart from being among the top 10 DEGs, were also identified as hub genes according to STRING analysis (Figure 2, Appendix A).

## 3. Discussion

The landscape of OS in ALL has evolved during the last decade, partially due to risk-adapted treatment regimens, including interventions like hematopoietic stem cell transplantation and targeted therapies, such as rituximab, imatinib, dasatinib, binatumomab, and inotuzumab [9]. Nevertheless, OS in adults with ALL exhibits distinct contrast across age groups and remains worse in Latin American populations. For instance, ALL adults from high-income countries achieve complete remission (CR) levels of 90%, with long-term OS of around 50% [22]. The current 5-year relative survival rates among Caucasians ages 20 to 49 years old are 47%, decreasing to 28% and 17% for age groups 50–64 and >65, respectively [23]. In Mexico, survival data from adults show improvement from 2.6 years (Disease evolution time: 2 months–15 years) in 2008 to 72.1% (two-year OS) in 2023 [12,24,25,26]. However, the 5-year OS is still low (43.7%), and data concerning adults and elderly adults are even worse (10.58 months). These studies have shown that risk stratification criteria and heterogeneous treatment protocols used across populations could explain low OS in low-income countries. Nevertheless, it has also been reported that there is a high prevalence of poor prognosis biomarkers such as Ph-like in Latin American populations [27,28]; thus, to increase our knowledge of ALL in adults, we performed a transcriptomic analysis using microarray gene expression. To the best of our knowledge, this is the first study reporting results of whole gene expression profiles in adults with ALL from Mexico.

### 3.1. Potential Biomarkers for Diagnosis and Prognosis

The comparative analysis among healthy subjects and patients revealed the elevated expression of markers of B cell development such as *DNTT*, *SOX4*, *EBF1*, as well as *MYB*, *CDK6*, and *EBF1*, etc. [29,30,31]. The most up-regulated gene was *DNTT* (TdT). This gene encodes TdT, which is required for the insertion of random nucleotides at VDJ joining regions during B- and T-cell receptor rearrangements; thus, its expression is restricted to normal and malignant pre-B and pre-T lymphocytes during early differentiation [29]. There are no data on the use of *DNTT* as a potential diagnostic biomarker; most of the studies have been conducted to know its usefulness as a prognostic biomarker. Based on an immunophenotypic test, it was reported that positivity for TdT expression is a prognostic factor in adults with ALL [32], which is in contrast to our results, as no association between *DNTT* and OS was detected. Conflicting data have also been reported for Acute Myeloid Leukemia (AML), where it has been suggested that *DNTT* expression is related to *FLT3-ITD* mutations [33] but not to survival [34]. However, it has also been reported that mutations in *DNTT* are associated with OS in AML [35], and different approaches could help decipher the clinical relevance of this gene in ALL.

Concerning *SOX4* and *EBF1*, it is known that both genes enable the survival signaling of leukemia cells [36,37]. Higher *SOX4* and *EBF1* expression levels in patients with ALL, in contrast to healthy subjects, forward their use as promising diagnostic biomarkers. It is notable that in a previous microarray expression analysis in Mexican children with ALL, we also observed a high expression of both genes in relapsed cases [21], and an association between the high expression of *SOX4* and worse prognosis was found.

*MYB* oncogene, which promotes uncontrolled neoplastic cell proliferation and blocked differentiation, has been found to be deregulated in ALL. Moreover, it has been suggested to be a potential prognostic marker and target for tailored therapy [38,39,40]. The levels of *MYB* and *CDK6* have been highly correlated in adults with Ph^+^. In fact, *CDK6* is a relevant target of *MYB*, having essential roles in the leukemogenesis of leukemic cells of this molecular subtype [41].

In relation to down-regulated genes, *PTGS2* and *PPBP* showed a very low expression in cases against controls. The overexpression of both genes has been associated with several human cancer types [42,43,44,45]. *PTGS2* or cyclooxygenase 2 (*COX2*) encodes prostaglandin–endoperoxide synthase, which is a relevant protein in oncogenic processes and has been shown to have a controversial association with ALL. According to our data, Vicent et al. [46] also observed no expression of *COX2* in blood samples taken from acute leukemia patients. However, in contrast to our findings, *COX2* was reported as being up-regulated in ALL, and data concerning cancer cell lines, including leukemia cells, have revealed that *COX2* inhibition reduces the growth of malignant cells [43,45]. Moreover, *COX2* has been suggested to be a potential target for therapeutic intervention to suppress pediatric ALL and improve OS [43,47]. It is well known that *COX2* is a pleiotropic protein, and no data from adults with ALL are available; thus, the relationship between *COX2* gene and ALL remains undeciphered.

*PPBP* has been involved in various cellular processes and malignancies [48,49,50,51]; in fact, it has been found to be down-regulated in the plasma of patients with gastric cancer (GC), and has been suggested to be a diagnostic biomarker of that disease [49,52]. Our combination of gene expression profiling with a computational approach using STRING showed *PPBP* to be a hub gene in ALL, as has been observed in AML, and based on its important role in biological processes, *PPBP* could be considered a potential drug target in acute leukemias [53,54].

Among the hub genes, *CD4* emerges as a key gene in adult ALL. This gene is expressed in peripheral blood monocytes, tissue macrophages, granulocytes, and helper/inducer T cells, and it is fundamental for T cell activation, thymic differentiation, and regulation of T-B cell adhesion [55]. In accordance with other studies relating to leukemia, we observed that the expression levels of *CD4* are a prognosis biomarker in ALL and other types of cancer.

### 3.2. Promising Therapeutic Target Genes

Among DEGs, *CDK6* and *FLT3* were found to be potentially targetable genes in adults with ALL. In fact, *FLT3* is already used as a treatment biomarker in AML [56,57], and data from cell lines showed that *FLT3L* CAR-T cells specifically kill FLT3+ leukemia [58]. Additionally, it holds potential in terms of monitoring Minimal Residual Disease (MRD) [59]. *ERG* is an Ets-transcription factor required for normal blood stem cell development [60], and the high expression of this gene has been associated with poor prognosis in AML [61]. It is noteworthy that *ERG* deletions have been found to occur recurrently in ALL, especially in the *DUX4*-rearranged subtype, and have a positive impact on the survival of ALL patients.

Regarding the ten top hub genes, *CD4*, *FCGR3A/FCGR3B*, and *TLR4* with plasma membrane location were identified as potential targeted therapies. Our data revealed that the high expression of *CD4* increases the risk of relapse and death (Figure 3). Recently, preclinical studies and a Phase I clinical trial reported that *CD4* CAR has cytotoxic effects against T cell malignancies, creating an opportunity to treat cases via the over-expression of this gene [62,63,64]. 

*FCGR3A*/*FCGR3B* emerged as a potentially druggable gene, predominates as a risk prognostic factor in most cancers, and is closely related to tumor immune-related pathways. More relevant, drug sensitivity analysis showed that higher *FCGR3A* expression predicts better efficacy to treatments based on antileukemic drugs such as Etoposide, Doxorubicin, and Methotrexate [65]. Additionally, IPA analysis suggested that *FCGR3A* is a target of trastuzumab, a monoclonal antibody proposed to treat ALL cases expressing the epidermal growth factor receptor *HER2/neu.* This receptor has been found to be overexpressed in around 30% of ALL patients and has been associated with chemoresistance and poor clinical outcomes in adults with this malignancy [66,67].

*TLR4* was down-regulated in our patients. The agonist of this gene has been a matter of many studies focused on cancer immunotherapy, and nowadays, FDA approval for clinical application in cancer treatment has been obtained for two *TLR4* ligands, Bacillus Calmette-Guérin and monophosphoryl lipid A [68,69].

Preliminary therapeutic target genes exploration represents a potentially relevant step toward improving treatment strategies for ALL patients but also raises the need for further investigation and validation of these genes. Comparing our results through the analysis of transcriptional datasets from adults with ALL across different ethnic groups could help to elucidate whether specific gene expression changes are associated with the disease or circumscribed to our population. However, the availability of microarray expression data of adults with ALL from other populations is deficient. Information such as that could potentially highlight novel population-specific gene expression changes in the Mexican ALL patients, which can be proposed for tailoring therapeutic designs and treatment regimens. Taking into account the valuable insights garnered from this study, it is essential to consider its inherent limitations, which provide a nuanced perspective on the findings and highlight avenues for future research refinement. On the one hand, the blood samples of the healthy subjects were used based on their accessibility and their biological relation with bone marrow, and although the data retrieved from Expression Atlas showed the same expression direction as our data, we cannot discard biases associated with intrinsic aspects of each tissue. On the other hand, the small sample size and the insufficient clinical and biological data of the patients limit conclusive results regarding the use of critical genes as biomarkers for diagnosis and prognosis. While the present findings have provided meaningful correlations and differential expressions, the inclusion of a larger, independent patient cohort would reinforce our results. Replicating the study with a more extensive cohort would offer a more comprehensive representation of the landscape of ALL in adults. Additionally, molecular data such as chromosomal abnormalities, gene mutations, MRD, etc., would contribute to identifying new prognostic biomarkers and potentially targeted molecules with treatment relevance. 

## 4. Materials and Methods

### 4.1. Biological Samples and Clinical Data Collection

Bone marrow (BM) samples of adults with de novo ALL were collected at the time of diagnosis in the period between July 2018 and September 2021. Cases with Down syndrome were excluded. All patients were recruited in the Hospital General de Mexico (HGM), a health institution located in Mexico City, which attends to cases from different regions of the country. ALL diagnosis was established by a hematologist based on the morphologic and immunophenotype features of leukemia cells. In addition, five blood samples obtained from healthy subjects were included. Clinical and demographic data collected at diagnoses such as gender, age, percentage of leukemic blasts in BM, immunophenotype, as well as initial treatment response, relapse, follow-up duration, and survival status were obtained from the patient’s clinical charts. Written informed consent was obtained from all participants.

### 4.2. RNA Extraction

For RNA extraction, white blood cells from the BM of patients and peripheral blood of healthy subjects were treated with Trizol reagent (Invitrogen Life Technologies, Carlsbad, CA, USA) and stored at −80 °C until their use. Cryopreserved samples were rapidly thawed, and total RNA was extracted and purified using standard protocols.

### 4.3. Gene Expression Microarrays Preparation

RNA was extracted from the BM and peripheral blood of patients with ALL and healthy subjects, respectively. The extracted RNA underwent evaluation via capillary electrophoresis using the Agilent bioanalyzer 2100 (Agilent Technologies, Santa Clara, CA, USA) to determine RNA integrity number (RIN), and values > 6.0 were included in the microarray analysis. The GeneChip Human Transcriptome Array 2.0 (HTA 2.0, Affymetrix Inc.; Santa Clara, CA, USA) interrogates both mRNA and lncRNAs. The GeneChip WT Plus Reagent Kit was used for the preparation of the microarray chips. The detailed protocol for sample preparation and microarray processing is available from Affymetrix. In brief, the first-strand cDNA was synthesized from 200 ng of RNA by employing SuperscriptII reverse transcriptase primed with a poly (T) oligomer, which incorporates the T7 promoter. Next, via in vitro transcription, the cRNA was obtained through the cDNA. This cRNA was used as a template for a second cDNA synthesis cycle with the incorporation of dUTPs into the new strand. After, the cDNA was fragmented through uracil-DNAc and purine–pyrimidine endonuclease. The obtained fragments were biotin-labeled (hybridization 45 °C for 16 h), stained, (streptavidin–phycoerythrin conjugate) washed, and scanned following Affymetrix HTA 2.0 chips protocols (Affymetrix Inc, Santa Clara, CA). A visual inspection to detect irregularities and for the data normalization of all Gene Chips was carried out. Quality measures, such as the percentage of present genes and the ratio of endogenous genes, indicated the high overall quality of the samples and assays. Scanning and data extraction of the microarrays was followed by the transformation of fluorescence data into CEL files employing the Affymetrix GeneChip Command Console (AGCC) version 4.0.0.1567 software.

### 4.4. Gene Expression Profiling Analyses

Microarray gene expression data were processed with the Affymetrix Transcriptome Analysis Console (TAC) v4.0.3 software. Background correction, probe set signal integration, and quantile normalization were performed through the Robust Multichip Analysis (RMA) algorithm, which was implemented through the use of Affymetrix Expression Console (ECS) v1.4 software [70]. To identify DEGs from healthy and tumoral tissues, supervised clustering analysis of gene expression was performed. Genes whose fold-change (FC) between each comparative group was >2.0 or ≤−2.0, with a *p* value cut-off of 0.05 and a false discovery rate (FDR) of < 0.05 was selected [71]. Probes with unassigned genes were discarded.

### 4.5. Pathway Enrichment Analyses of Differentially Expressed Genes and Protein–Protein Interaction Network

The Database for Annotation, Visualization, and Integrated Discovery (DAVID) tool was used to evaluate the functional annotation and enrichment analysis of DEGs (https://david.ncifcrf.gov, accessed on 13 November 2023) [72]. The gene enrichment pathway analysis of DEGs and the identification of potential therapeutic target genes were performed through the use of Ingenuity Pathway Analysis (IPA, QIAGEN Redwood City, www.qiagen.com/ingenuity, accessed on 6 November 2023.) Protein–protein interaction (PPI) was evaluated by using the STRING database (https://string-db.org, accessed on 13 November 2023) by using a high order of confidence (>0.07) [73], and a gene–gene interaction network was constructed and visualized by using Cytoscape (v3.10.1) [74], and a *p* value < 0.05 was considered statically significant.

### 4.6. Quantitative Real-Time PCR for Microarray Data Validation

By using real-time quantitative reverse transcription PCR (q-RT-PCR), two candidate genes (FC > 2, FDR < 0.05) were selected to validate the microarray expression results as we have described previously [21]. Reactions were carried out using standard protocols. Briefly, cDNA was synthesized from 250 ng of total RNA for each sample using OdT primer and the High-Capacity cDNA Reverse Transcriptions Kit. cDNA reactions were performed in a final volume of 20 μL under the following conditions: at 25 °C for 25 min, at 37 °C for 120 min, and at 85 °C for 5 min in a GeneAmp PCR System 9700 (Applied Biosystems). For quantitative purposes, we used the SYBR Select Master Mix (Applied Biosystems, Carlsbad, CA, USA) method, and PCR was performed in a QuantStudio^TM^ 3 Real-Time PCR system with the following conditions: one cycle at 50 °C for 2 min, at 95 °C for 15 s, at 54.8 °C for 15 s, and 72 °C for 1 min, for a total of 40 cycles. FC was calculated using the 2^−DDCt^ method [75]. Primers were designed using the Primer-BLAST Tool (https://www.ncbi.nlm.nih.gov/tools/primer-blast/, accessed on 13 November 2023) [76] (Appendix A), and the ACTB gene was used for data normalization.

### 4.7. Gene Expression Correlation Analysis

Gene–gene expression correlation analysis was performed by using the expression data from microarray experiments that were preprocessed using standard normalization tools (log transformation and quantile normalization) to ensure comparability and reduce technical variations. Subsequently, pairwise correlation coefficients were calculated to assess the strength and direction of the association between the expression profiles of individual genes. To identify significant correlations, statistical tests of Spearman’s rank correlation coefficient were applied, and a *p*-value threshold of 0.05 was used to determine the statistical significance of the correlations. To account for multiple tests, a correction method, such as the Benjamini–Hochberg procedure, was applied to control the FDR.

### 4.8. Statistical Analysis

Comparisons of demographic and clinical variables across groups were made using Chi-square, *X*^2^, or Fisher’s exact test for categorical data; *p*-values < 0.05 were considered statistically significant. According to the Shapiro–Wilk test (*p* < 0.05), gene expression data were not normally distributed; thus, the cut-off value was determined based on the median, and two groups were identified: low and high gene expression. Survival analysis was carried out by using the Kaplan–Meier method. Event-free survival (EFS) and overall survival (OS) were calculated for each of the top 20 DEGs based on the expression levels (high or low). OS was measured from the initial treatment date until the last follow-up date. The log-rank test was used to compare differences between survival curves; a *p*-value less than 0.05 was considered statistically significant. Adjusted hazard ratios (HRs) and corresponding 95% confidence intervals were calculated to assess the significance of these abnormally expressed genes.

## 5. Conclusions

The present work represents the first effort to identify DEGs that may be involved in leukemogenesis in Mexican adults and identify potential prognosis biomarkers. Our findings suggest that the expression level of *DNTT*, *MYB*, *EBF1*, *PTGS2*, and *PPBP*, among others, in blood samples could discriminate patients with ALL from healthy subjects. As well, the over-expression of *ERG*, *SOX4*, and *CDK6*, and the low expression of the hub gene *CD4* are significantly associated with inferior outcomes. Our study presents an important step to understanding the pathophysiology of ALL in adults, but the exact prognostic value of these critical genes requires detailed evaluation using clinical data from a larger cohort of patients with ALL recruited from multiple centers.

## Figures and Tables

**Figure 1 ijms-25-01750-f001:**
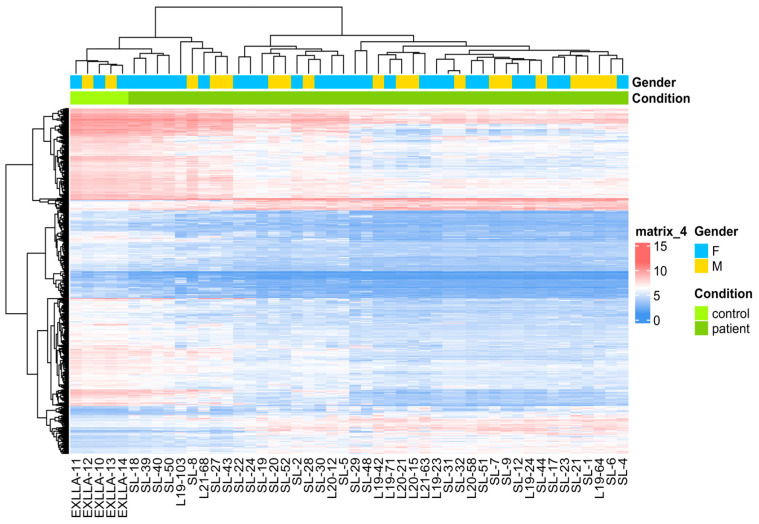
Analysis of gene expression profiles in acute lymphoblastic leukemia (ALL). Heat map shows the differential gene expression (DGE) patterns between ALL and non-ALL subjects. The heat map is constructed using 871 DGEs. The expression is reported through a color pattern, with red and blue for high and low levels of gene expression, respectively. Color intensity reflects the level of gene expression.

**Figure 2 ijms-25-01750-f002:**
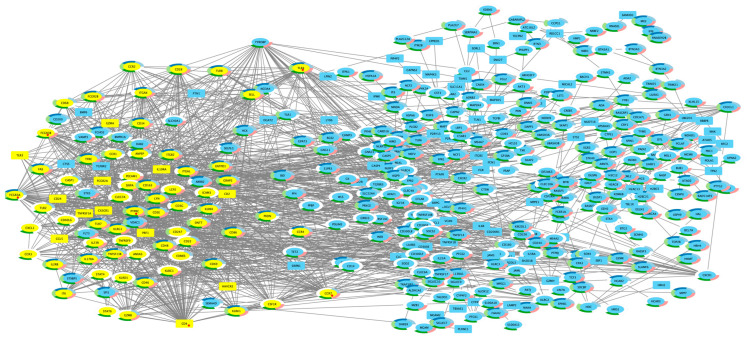
The figure shows the network model visualized using Cytoscape (v3.10.1). Nodes with yellow fill identify genes *CD4* primary interaction. The black line represents the PPI relationship between the nodes. Colors coded by biological processes: light blue (immune system process), dark blue (immune response), aqua-green (regulation of immune system process), green (response to stimulus), and pink (response to stress). PPI: protein–protein interaction. Red dot: most ranked nodes.

**Figure 3 ijms-25-01750-f003:**
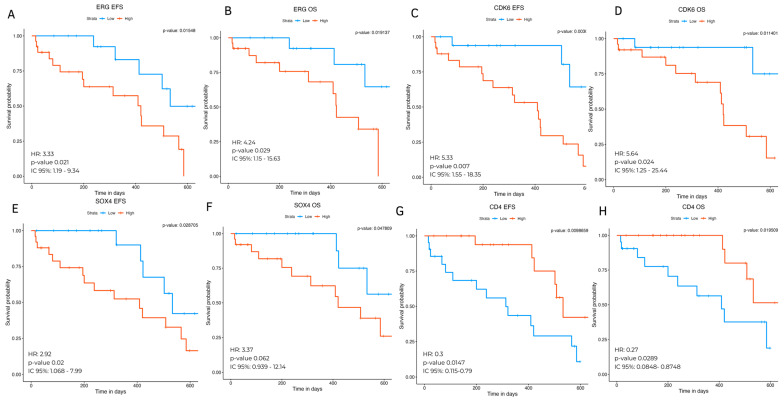
Event-free survival (EFS) and overall survival (OS) analyses of unregulated genes. Expression of *ERG* (**A**,**B**), CDK6 (**C**,**D**), *SOX4* (**E**,**F**) and *CD4* (**G**,**H**).

**Figure 4 ijms-25-01750-f004:**
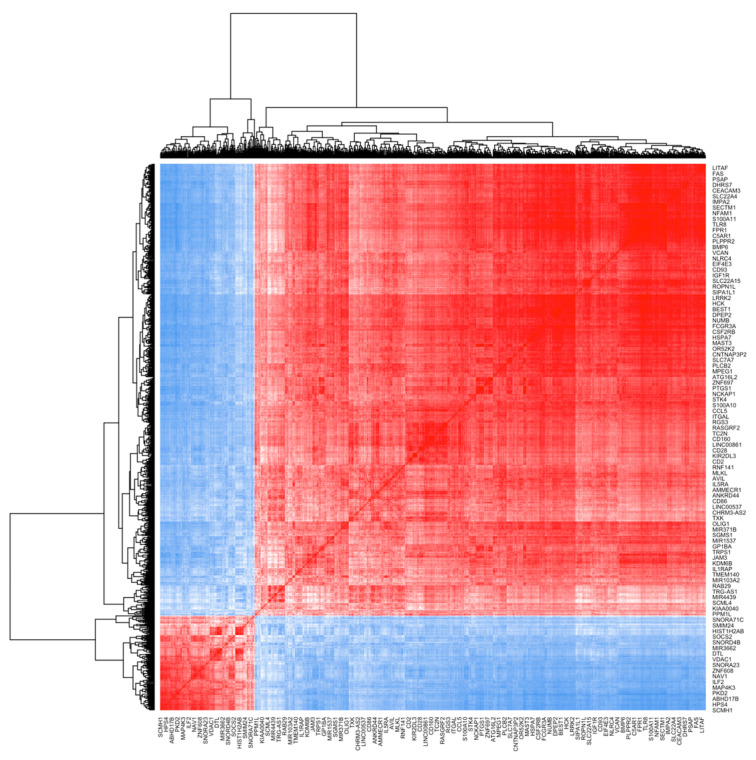
Heat map of the gene co-expression network. The heatmap describes the correlation among gene expression. Each row and column of the heatmap corresponds to a single gene. Blue: low correlation; red: high correlation. Color intensity reflects the level of gene correlation.

**Table 1 ijms-25-01750-t001:** Clinical characteristics of the patients with ALL included in this study.

Characteristic	*n* = 43	Percentage
Gender		
Male	23	53.5%
Female	20	46.5%
Median age at diagnosis in years: 33.8 (18–57)		
AYA (18–40) *	27	62.8%
>40 years	16	37.2%
Risk classification		
Standard	5	11.6%
High	38	88.4%
Relapse		
Yes	16	37.2%
No	27	62.8%
Immunophenotype		
B	37	86%
Pre-B	4	9.3%
T	2	4.7%
Death		
Yes	16	37.2%
No	27	62.8%

* AYA: adolescents and young adults.

**Table 2 ijms-25-01750-t002:** The top 10 up-regulated and top 10 down-regulated ranked differentially expressed genes in adult acute lymphoblastic leukemia patients compared to controls.

Gene Symbol	Fold Change	*p* Value	FDR ^1^
*DNTT*	107.25	4.44 × 10^−5^	0.0024
*MYB*	39.05	3.18 × 10^−7^	5.19 × 10^−5^
*SOX4*	19.52	2.49 × 10^−5^	0.0015
*EBF1*	19.11	5 × 10^−4^	0.0153
*ERG*	14.7	2.76 × 10^−6^	3 × 10^−4^
*CD34*	9.33	0.0019	0.0372
*FLT3*	11.55	2 × 10^−4^	0.0069
*STMN1*	8.41	6.29 × 10^−5^	0.0032
*CDK6*	8.07	9.75 × 10^−6^	8 × 10^−4^
*NAV1*	7.24	0.0013	0.0284
*SH3BGRL2*	−20.18	1.4 × 10^−6^	2 × 10^−4^
*CLEC7A*	−20.91	1.67 × 10^−5^	0.0011
*RGS2*	−21.07	4.46 × 10^−6^	4 × 10^−4^
*PF4*	−23.09	1.11 × 10^−5^	8 × 10^−4^
*TUBB1*	−24.7	8.38 × 10^−7^	1 × 10^−4^
*VCAN*	−25.22	3.23 × 10^−5^	0.0019
*LUCAT1*	−27.02	6.62 × 10^−9^	2.05 × 10^−6^
*ADGRE3*	−27.11	2.04 × 10^−7^	3.6 × 10^−5^
*PPBP*	−52.24	8.79 × 10^−5^	0.0041
*PTGS2*	−57.83	8.5 × 10^−8^	1.76 × 10^−5^

^1^ FDR: False discovery rate.

## Data Availability

The expression data are available upon request to the corresponding author. The data are not publicly available due to further analyses are underway.

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
