# Peer review of "Transcriptome Analysis in Mexican Adults with Acute Lymphoblastic Leukemia"

_ijms, 2024, doi:10.3390/ijms25031750_

Round 1
Reviewer 1 Report
Comments and Suggestions for Authors
In this work, the investigators analyzed the transcriptional profiles of adult ALL patients, and identified hub genes and biological processes associated with gene expression changes. The study is limited in scope, but the authors have been clear in explaining the rationale of the study, and the results. I would recommend that the authors analyze publicly available transcriptional datasets of adult ALL patients across different races and ethnic groups, and compare the results with their dataset to elucidate specific gene expression changes in the sample cohort they have analyzed. Information such as that could potentially highlight novel population specific gene expression changes in the Mexican ALL patients which can they be proposed for tailoring therapeutic designs and treatment regimens.
Comments on the Quality of English Language
There are typographical and grammatical errors in the manuscript. Please edit the manuscript thoroughly and correct those.
Author Response
Dear Reviewer 1,
We sincerely appreciate the time and effort that you have dedicated to providing us your valuable feedback on our manuscript. We are pleased that you found the exposition of the study and its results clear. We have carefully addressed each of your points and changes within the manuscript have been highlighted in bold underlined words.
Comment. 1. I would recommend that the authors analyze publicly available transcriptional datasets of adult ALL patients across different races and ethnic groups, and compare the results with their dataset to elucidate specific gene expression changes in the sample cohort they have analyzed.
Response: We are totally agree, it would have been interesting to explore this. However, we were limited to do it, because transcriptomic studies of ALL have been mainly focused on pediatric populations and we could not find available microarray transcriptomics datasets from adult patients. The Therapeutically Applicable Research to Generate Effective Treatments (TARGET) initiative is a public repository of RNASeq data of 463 patients with ALL ranged 2-30 years old of age (<5% of them are adults) from Hispanic and non-Hispanic patients (https://ocg.cancer.gov/programs/target). Despite ages differences and expression analysis approaches among TARGET and our Mexican cohorts, we analyzed three up-regulated genes (MYB, SOX4, and ERG) and two down-regulated genes (PPBP and PTGS2), which were chosen from table 1 (main manuscript). TARGET cohort was compared with The Genotype-Tissue Expression (GTEx) cohort (407 non-cancerous patients, https://gtexportal.org/home/) using the TNMplot platform [1]. Findings were similar to our results. A figure was included for your consideration (Supporting file).
Additionally, we added the following statement in the manuscript (Discussion section, line 241): Comparing our results through the analysis of transcriptional datasets from adult with ALL across different ethnic groups could help elucidate whether specific gene expression changes are associated with the disease or circumscribed to our population. However, the availability of microarrays expression data of adults with ALL from other populations is deficient.
Reference
1. Bartha Á, Győrffy B. TNMplot.com: A Web Tool for the Comparison of Gene Expression in Normal, Tumor and Metastatic Tissues. Int J Mol Sci (2021) 22(5):2622. doi: 10.3390/ijms22052622
2. Comment 2. There are typographical and grammatical errors in the manuscript. Please edit the manuscript throughly and correct those.
Response: We appreciate your valuable comments, the manuscript has been revised and corrected.
We appreciate your constructive and insightful comments, we hope, we have addressed all your concerns.
Sincerely,
Silvia Jiménez Morales, PhD.
Corresponding author

Reviewer 2 Report
Comments and Suggestions for Authors
The manuscript covers potential therapeutic targets for ALL treatment. However, the selected DEGs have to be further verified.
1. Provide ethics approval documents
2. At line 316, ‘RNA extracted from BM patients’ is not clear and could mean that there is disease called BM and RNA was extracted from patients with BM.
3. Some of DEGs are high p value and FDR on table 2 and some of those markers were used for further analysis. This weakens the rest of figures and discussion as it may not be DEGs in ALL
4. Figure 2 resolution is poor. Cannot find or read any markers
5. It is very obvious that there are distinctive two groups in patient samples on Supplementary figure 2. Patient can be grouped by high FLT3 and low. This implies again that FLT3 may not be a proper DEG (on table 2, FLT3 p value is 2).
Comments on the Quality of English LanguageThere is no major issues with the quality of English language
Author Response
Dear Reviewer 2,
Thank you for your insightful feedback on our manuscript, your valuable suggestions helped us to substantially improve it. We have carefully addressed each of your points. Please find the detailed responses below and the corresponding revision highlighted in bold underlined words in the re-submitted files.
- Commments 1. Provide ethics approval documents
Response 1: We have already included this information in the "Institutional Review Board Statement" section, line 370 as follow: “... and approved by the Institutional Ethic Committee of Hospital General de México (number: D1/16/103/03/035).
- Comments 2: At line 316, ‘RNA extracted from BM patients’ is not clear and could mean that there is disease called BM and RNA was extracted from patients with BM.
Response 2: Accordingly, we have changed (line 275) the statement as follow: “RNA was extracted from BM and peripheral blood of patients with ALL and healthy subjects, respectively”.
- Comments 3: Some of DEGs are high p value and FDR on table 2 and some of those markers were used for further analysis. This weakens the rest of figures and discussion as it may not be DEGs in ALL
Response 3: We appreciate your comment. Table 2 was corrected and the missing information has been added.
- Comments 4: Figure 2 resolution is poor. Cannot find or read any markers Response 4: Thank you for your suggestions, the resolution of figure 2 has been improved. In addition, we included a .ppt file to ensure the quality of the image.
- Comments 5. It is very obvious that there are distinctive two groups in patient samples on Supplementary figure 2. Patient can be grouped by high FLT3 and low. This implies again that FLT3 may not be a proper DEG (on table 2, FLT3 p value is 2).
Response 5: Thank you very much for pointing this out. We have corrected table 2 to clarify this point. Additionally, we agree that Supplementary figure 2 suggests that there are distinctive two groups in our studied population. However, data presented in that figure was obtained from validation analysis of the microarray expression results using qRT–PCR on a selected subset of patients from the discovery cohort, which could explain the presence of two high and low FLT3 expression groups. To support this idea, we reviewed the FLT3 expression derived from our microarray analysis, including all cases. Differences were observed between cases and controls, but we did not identify two groups defined by high or low FLT3 expression (Figure A).
Additionally, we retrieved RNASeq data from The Therapeutically Applicable Research to Generate Effective Treatments (TARGET) initiative. This data base is a public repository of RNASeq data of 463 patients with ALL ranged 2-30 years old of age (<5% of them are adults) from Hispanic and non-Hispanic patients (https://ocg.cancer.gov/programs/target). We compared the expression of FLT3 between TARGET cohort and The Genotype-Tissue
Expression (GTEx) cohort (407 non-cancerous patients, https://gtexportal.org/home/) using the TNMplot platform (figure B)[1]. Despite ages differences and expression analysis approaches among TARGET and our Mexican cohorts, findings were similar to our results. Figures A and B were included for your consideration as a supporting file.
In the Validation of DEGs associated with ALL by Quantitative RT-PCR section (line 130), we added the followed statement: "...in a subset of cases and controls..."
We hope these clarifications and revisions address all your concerns.
Sincerely,
Silvia Jiménez-Morales, PhD.
Corresponding author

Round 2
Reviewer 2 Report
Comments and Suggestions for Authors
The manuscript needs to be revised after accepting all the editions. For example, line 164 - 165, where is 'in a subset of cases and controls.' supposed to be for?
To investigate FLT3 high and low samples, authors need to try different statistical analysis such as principal component analysis and others. If additional statistical analysis still show the same results then supplementary figure data is true
Comments on the Quality of English LanguageStill need to go revision one more round
Author Response
Dear Reviewer,
We sincerely appreciate the time and effort that you have dedicated to providing us your valuable feedback on our manuscript. We have carefully addressed each of your points and changes within the manuscript have been highlighted in bold underlined words.
Comment. 1. The manuscript needs to be revised after accepting all the editions. For example, line 164 - 165, where is 'in a subset of cases and controls.' supposed to be for?
Response: We appreciate your valuable comments, the manuscript has been revised and corrected.
2. Comment 2. To investigate FLT3 high and low samples, authors need to try different statistical analysis such as principal component analysis and others. If additional statistical analysis still show the same results then supplementary figure data is true
Response: According your valuable suggestions, we conducted a principal component analysis (PCA) base on FLT3 expression. Since PCA showed inconclusive results (Figure 1, supporting file), we performed a Shapiro-Wilk test to evaluate normal distribution of FLT3 expression and a Levene's test for assessing the homogeneity of variance between high and low expression groups. Both analyses support our previous findings (there is a wide range of FLT3 expression among adults with ALL) suggesting the absence of distinct groups within FLT3 expression in the patient cohort (Figure 2 and 3, supporting file). Therefore, we have considered that the distinctive two groups observed in patient samples on Supplementary figure 2 are due to biases during the election of the samples to be tested by RT-PCR during the validation of the microarrays expression data. To let you know, we had very limited RNA quantity from some leukemia cases, thus it was only feasible to conduct the validation test in a subgroup o the cases. To note the potential bias during our validation analysis, Supplementary figure was modified according this: Validation of the differential expression analysis results for the genes EBF1 and FLT3 using double delta Ct method in a subset of patients.
Further, your notable observation has forces us to deepen into the analysis of our cases and we detected that mostly patients located in the right side of the PCA plot (red dots, figure 1) are female, and a recent study has shown sex disparity in acute myeloid leukaemia with FLT3 mutations (Hellesøy et al 2021). Thus, we have to explore this interesting gene in future researches.
Finally, we appreciate your diligence in reviewing our work, and we hope that this version have addressed all your concerns. Your valuable suggestions have enhanced the quality of our manuscript and we remain open to any additional suggestions or comments you may have.
Sincerely,
Silvia Jiménez Morales, PhD.
Corresponding author

Round 3
Reviewer 2 Report
Comments and Suggestions for Authors
Authors answered all the comments properly.